# Identification and Profiling Analysis of microRNAs in Guava Fruit (*Psidium guajava* L.) and Their Role during Ripening

**DOI:** 10.3390/genes14112029

**Published:** 2023-10-31

**Authors:** Mario Alejandro Mejía-Mendoza, Cristina Garcidueñas-Piña, Blanca Estela Barrera-Figueroa, José Francisco Morales-Domínguez

**Affiliations:** 1Departamento de Química, Centro de Ciencias Básicas, Universidad Autónoma de Aguascalientes (UAA), Av. Universidad, #940, Ciudad Universitaria, Aguascalientes 20100, Mexico; mendozamario7@hotmail.com (M.A.M.-M.); cristina.garciduenas@edu.uaa.mx (C.G.-P.); 2Centro de Investigaciones Científicas, Laboratorio de Biotecnología Vegetal, Instituto de Biotecnología, Universidad del Papaloapan, Circuito Central #200, Parque Industrial, Tuxtepec 68301, Mexico; bbarrera@unpa.edu.mx

**Keywords:** climacteric fruit, ripening, gene regulation, transcription factors, microRNAs, guava

## Abstract

The guava (*Psidium guajava* L.) is a climacteric fruit with an accelerated post-harvest overripening. miRNAs are small RNA sequences that function as gene regulators in eukaryotes and are essential for their survival and development. In this study, miRNA libraries were constructed, sequenced and analyzed from the breaker and ripe stages of guava fruit cv. Siglo XXI. One hundred and seventy-four mature miRNA sequences from 28 miRNA families were identified. The taxonomic distribution of the guava miRNAs showed a high level of conservation among the dicotyledonous plants. Most of the predicted miRNA target genes were transcription factors and genes involved in the metabolism of phytohormones such as abscisic acid, auxins, and ethylene, as revealed through an ontology enrichment analysis. The miRNA families miR168, miR169, miR396, miR397, and miR482 were classified as being directly associated with maturation, whereas the miRNA families miR160, miR165, miR167, miR3930, miR395, miR398, and miR535 were classified as being indirectly associated. With this study, we intended to increase our knowledge and understanding of the regulatory process involved in the ripening process, thereby providing valuable information for future research on the ripening of guava fruit.

## 1. Introduction

The guava (*Psidium guajava* L.) is a fruit of global economic and social significance due to its high content of compounds with pharmacological activity and nutritional contribution. In Mexico, for more than a century, the guava has been a crop of high commercial interest due to a large number of derivative products such as jellies, candies, and some anti-inflammatory stomach medications. As the guava is a climacteric fruit, its maturation is determined by its ethylene production levels [1]. The ripening process is a series of physical, chemical, and biochemical changes that occur in the fruit due to the joint and coordinated action of enzymes, coding genes, and transcription factors [2]. During this process, gene regulation is mediated by various types of RNA, including microRNAs (miRNAs) [3].

miRNAs are small non-coding RNA sequences, with between 18 and 22 nucleotides, that regulate the expression of genes at the transcriptional and post-transcriptional levels in eukaryotes. miRNAs are generated in the cell nucleus from RNA transcripts called pri-miRNAs, which originate hairpin-shaped structures or loops, called pre-miRNAs, capable of entering the cytoplasm and producing functional or mature miRNAs [4]. miRNAs act by binding to complementary messenger RNA (mRNA) sequences or targets [5]. With these findings, interest in the study of miRNAs has arisen due to their ability to regulate gene expression [6], as they play crucial roles in essential functions for the proper growth and development of plants, including the organ formation and differentiation, flowering, and reproduction processes [7]; phytohormone signaling; plant communication [8]; responses to biotic and abiotic stresses [9]; and responses to programmed cell death [10], among others.

Recent studies have shown that miRNAs also play a role in fruit ripening by regulating genes and transcription factors such as GRAS *(GIBBERELLIN-INSENSITIVE (GAI)*, *REPRESSOR OF GA1-3 (RGA)*, and *SCARECROW (SCR)),* which have roles in ethylene biosynthesis during the earliest stages of ripening [11]; the enzyme β-galactosidase, which participates in the modification of the cell wall and therefore in the softening of the fruit [12]; the gibberellin-dependent MYB (GAMYB) transcription factors, which perform functions of cell wall modification and ABA signaling during ripening [13]; auxin response factors (ARFs) that act on the metabolic pathways of ethylene signaling [14]; and the auxin transport inhibition/response F-Box proteins (TIR1/AFBs) TIR1/AFB2, which participate in the ubiquitin degradation of indole acetic acid (IAA) and auxins, releasing ARF proteins [3]. These investigations relied on the construction and comparative analysis of plant miRNA libraries [3,12,15].

Currently, there are no studies on miRNAs associated with the ripening of Mexican guava fruit. Therefore, in this study, the characterization and identification of the miRNAs found in the guava fruit Media China group, cv. Calvillo Siglo XXI at the breaker and ripe stages, as well as their taxonomic distribution in other plants, the metabolic processes affected by the regulation of their target genes, and their ontological enrichment, were conducted. This research aimed to expand our understanding of the genes involved in the ripening of guava fruit and the miRNAs that regulate their expression.

## 2. Materials and Methods

### 2.1. Plant Material

Guava fruits (*P. guajava*, “MediaChina” group, cv. Siglo XXI) at the breaker (95 days after flowering) and ripe (125 days after flowering) stages were collected from a single tree established in the Germplasm Bank of the Los Cañones Experimental Site, belonging to INIFAP, which is located in the municipality of Huanusco, Zacatecas, Mexico (latitude: 21°44′43.6″ N; longitude: 102°58′02.0″ W).

### 2.2. Total RNA Extraction and Small RNA Library Construction

The total RNA was extracted from only the pericarp of the guava fruits at the breaker and ripe stage (three replicates each). The pericarp was frozen with liquid nitrogen and ground for RNA extraction using the López-Gómez and Gómez-Lim methods [16]. After the loading of 12 µg of total RNA per sample in a 15% denaturing acrylamide gel, the 20–27 nt small RNA fraction was separated and purified. The small RNAs were then ligated at their 3′ ends to a pre-adenylated adapter (5′-AppCTGTAGGCACCATCAAT-NH2-3′) and at their 5′ ends to the RNA adapter Illumina RA5 (5′-GUUCAGAGUUCUACAGUCCGACGAUC-3′). cDNA synthesis was carried out with the Maxima First Strand cDNA Synthesis Kit (ThermoScientific, Waltham, MA, USA). The six cDNA libraries were amplified in a 15-cycle PCR reaction with Phusion High-Fidelity DNA polymerase (New England Biolabs, Ipswich, MA, USA), using various index primers from the Illumina small RNA index primer kit. The libraries were purified and sequenced using an Illumina Next-Seq 500 sequencer at the Biotechnology Institute of the National University of Mexico in Cuernavaca, Morelos.

### 2.3. Bioinformatic Analysis of miRNA Libraries

The small RNA libraries were divided into two groups, the ripe stage and the breaker stage, each with three replicates, for a total of six libraries. Each repetition was sequenced in duplicate, resulting in a total of 12 datasets. After sequencing, the raw reads were analyzed and processed using QIAGEN CLC Genomics Workbench 12.0. software (QIAGEN, Hilden, Germany). Clean sequences were obtained by removing adapter sequences, low-quality reads, and sequences below 17 nucleotides (nt) or higher than 25 nt. The clean sequences were mapped against the complete genome of *P. guajava* (assembly guava_v11.23) (GCA_016432845.1) from the NCBI database (https://www.ncbi.nlm.nih.gov/ accessed on 10 June 2019) using the “Maps read to reference” function of the CLC Genomics Workbench. Sequences matching non-coding RNAs (tRNAs and rRNAs, among others) were removed from the data. The remaining sequences were mapped against the miRNA database (Version 21) from the miRBase (http://mirbase.org/ accessed on 15 June 2019) to find conserved miRNAs, allowing (1) zero mismatches; (2) up to two mismatches to detect polymorphisms; or (3) the containment of additional external bases matching to the precursor (sup), missing external bases matching to a mature sequence (sub), or both (sub/sup) to find the positional variants of mature miRNAs (VS/Ss). Finally, with the Annotate and Merge Counts function of CLC Genomics Workbench, the sequences were grouped based on frequency of reads and normalized to one million of the total number of miRNA readings in each sample. Then, differential expression analysis of individual miRNA sequences was performed. Log2 (normalized counts in ripe stage/normalized counts in breaker stage) was used to compute fold change values. For differential expression analysis, only miRNA sequences with a raw count above 10 in at least one of the libraries were considered, and fold change values above 1.0 and below −0.1 were deemed upregulated and downregulated, respectively. Ontological enrichment analysis of the miRNA target genes was performed using the R statistical package [17], the PANTHER database, and the REVIGO software (Revigo version 1.8.1) tool to define the biological processes of the miRNA target genes.

### 2.4. Availability of Data and Materials

The miRNA-seq raw sequencing data have been deposited in the database of the National Center for Biotechnology Information (NCBI) and can be accessed using the accession number PRJNA996171.

## 3. Results and Discussion

### 3.1. Profiling of miRNA Families Found in Guava Fruit

The six miRNA libraries from the ripe and breaker stages of the fruit were cleaned and analyzed for size distribution, isoforms, sub and sup variants, taxonomic distribution, and functionality during the ripening process. There was a total of 16,729,248 readings obtained. After removal of the adapters and filtering of the sequences, 244,277 sequences between 18 and 25 nt were retained. After comparison to the miRBase database, only 630 sequences remained. Only 3’ and 5’ mature miRNAs with VS/Ss were chosen, yielding 174 sequences (Appendix A). The 174 sequences were categorized into 28 miRNA families (FMIRs), along with their respective isoforms and the VS/Ss of each family (Figure 1 and Table 1). FMIR166, with 12 isoforms and 43 variants, was the family with the greatest number of isoforms and VS/Ss, followed by FMIR319, FMIR159, and FMIR393. FMIR159 (29), FMIR319 (14), FMIR482 (13), and FMIR168 (12) were the families with the highest VS/S numbers.

The size of the miRNAs ranged between 18 and 22 nucleotides, with 21 nucleotides being the most prevalent (75 sequences), followed by 20 nucleotides (35 sequences). These ranges and sizes are supported by the majority of the plant-miRNA-related literature [12,15,18] (Appendix A).

### 3.2. Evolutionary Divergence and Conservation Analysis of P. guajava microRNAs

FMIR166 is the most conserved gene among species such as *Arabidopsis thaliana* and *Oryza sativa* [6,9], as indicated by the taxonomic distribution (Figure 2). In addition to being detected in guava fruit, FMIR167 and FMIR319 have been identified in *Cucumis melo* [3]. The *Solanaceae* family contains fourteen highly conserved FMIRs in guava fruit, followed by *Myrtaceae* and *Eugenia uniflora*, with ten FMIRs. Ten other families are also present in *Brassicaceae*, *Euphorbiaceae*, *Linaceae*, and *Salicaceae* from the *Malphigiales* order. The FMIR156, FMIR164, FMIR166, FMIR167, FMIR168, FMIR169, FMIR171, FMIR319, FMIR394, FMIR395, FMIR396, and FMIR535 found in guava fruit have also been identified and expressed in climacteric and non-climacteric fruits such as apples, melons, tomatoes, bananas, and kiwis, among others [2,3,12,18], demonstrating that FMIR156, FMIR159, FMIR171, and mainly FMIR319 are present in all the studied fruits during ripening [19].

### 3.3. MiRNA Families Expressed in Guava Fruit and Their Target Genes

Due to their high numbers of VS/Ss, isoforms, and fold change values, FMIR159, FMIR165, FMIR166, FMIR167, FMIR168, FMIR169, FMIR171, FMIR319, FMIR393, FMIR395, FMIR396, FMIR397, FMIR398, FMIR482, and FMIR535 stand out among the 28 FMIRs in guava fruit (Appendix A). On the basis of recent investigations of the function of miRNAs during fruit ripening in melons, “Kyoho” grapes, bananas, blueberries, tomatoes, and kiwis [3,11,12,14,18,19], it is hypothesized that these miRNAs may have a similar function in guava fruit, as well as the same target genes (Table 1).

In both the breaker and mature stages of guava fruit, FMIR159 a/b/c isoforms were identified, with isoform 159b being the most abundantly expressed, followed by isoforms 159a and 159c. These findings are consistent with those observed in A. thaliana mutants [6] deficient in miRNA159a/b/c, where isoforms 159a/b were found to be the most abundantly expressed. FMIR159 is one of the most conserved miRNAs in both monocotyledonous and dicotyledonous plants, and its expression has been observed in all organs and tissues [6,20]. The transcription factors *GAMYB33*, *GAMYB65*, *Fa-GAMYB*, and *SIGAMYB* are the target genes of FMIR159 [6,21,22]; their functions include anther generation [21], the promotion of ripening through the regulation of gibberellin metabolism [23], signaling by ABA accumulation [11], and callose transport in early fruit development stages [13]. The inhibition of these transcription factors by FMIR159 in strawberries (*Fragaria vesca*) and tomatoes (*Solanum lycopersicum*) has been shown to have positive effects on the ripening process through the formation of the receptacle, fruit coloration, ovule production, and early development [21,22].

From FMIR165, the 165a/b isoforms were expressed in both the breaker and ripe stages of guava fruit. Its target genes are the transcription factor *Homeodomain-Leucine Zipper III (HD-Zip III)* [24,25,26] and the factors *PHB, REVOLUTA (REV)* and *PHV*, which are involved in the biosynthesis, signaling, and transport of auxins by promoting the expression of the genes *TRYPTOPHAN AMINOTRANSFERASE OF ARABIDOPSIS1 (TAA1)*, *YUCCA5 (YUC5)*, *LIKE AUXIN RESISTANT2 (LAX2)*, and *LAX3*, respectively, which are essential in the development, growth, and ripening processes [24,25].

Isoforms a/b/c/d/e/f/h/i/k/m/p/u of FMIR166 were found in both ripening stages of guava fruit; this FMIR also had the highest number of VS/Ss at forty-three. Similar to FMIR165, this family modulates multiple development parameters. However, its role in maturation can be divided into two functions; the first is its affinity for HD-Zip III proteins, which regulates processes such as the apical growth of meristems, the induction of lateral organs, and root formation through the regulation of auxin transport in the plant [27]. The second is involved in the unpacking and modification of the cell wall through the inhibition of genes related to the biosynthesis of residues of glycosphingolipids and galactose, which are key components of the cell wall and whose modification or degradation is required for the softening of the fruit [12,28,29].

FMIR167 a/f isoforms were only identified in the breaker stage, whereas isoforms b/c were only detected in the mature stage. In experiments with Cavendish banana, melon, and grapefruit (*Citrus grandis*) fruits, FMIR167 expression was observed during the ripe stage, corresponding to its expression profile in guava fruit [12,30,31]. FMIR167 targets the *ARF6* and *8* and *IAR3* genes, which encode auxin-conjugate amidohydrolase proteins, which aid in the conjugation/deconjugation of auxins with amino acids. These genes are essential for anther dehiscence, ovule development, fertility, and auxin metabolism, indicating an association with ripening [3,32]; therefore, they may play a similar function in guava fruit.

The a/b isoforms of FMIR168 were detected in both phases of guava ripening. In the ripe stage, eight distinct VS/Ss were observed, whereas only five were discovered in the breaker stage. The expression of this FMIR during the ripening of climacteric fruits has been previously described in guava fruit throughout all phases of ripening, with the highest expression occurring in the final stages [18,33]. FMIR168 targets genes that encode the *AGO1* protein, a fundamental component of the RNA-induced silencing complex (RISC) that serves as an anchor protein for microRNAs [33,34]. *AGO1* is regulated post-transcriptionally by miRNA168 through cleavage [35]. Its overexpression in tomatoes has led to the accumulation of its own miRNA as well as miR156, miR172, miR166, miR171, and miR393 in cytosol [34], supporting the notion that it functions as a maturation-related regulator of other miRNAs.

FMIR169 isoforms a/g were found in both stages of maturation, in accordance with the expression profiles in climacteric fruits [18]. Studies of tomatoes have shown that overexpression of this FMIR affects the coloration of ripening fruits due to the absence of NF-YA activity [36].

FMIR171 only expressed the isoform miR171f when the guava fruit was mature. This isoform was also detected in the final phases of ripening of strawberry (*F. vesca*) fruit, and it was found to be overexpressed in the presence of high CO_2_ concentrations [37]. miR171 controls GRAS factors, whose function centers on gibberellic-acid-mediated signaling/repression and root and stem development [38]. Through BLASTn and multiple alignment analyses, we found binding sites of miR171f in the *GRAS4*, *GRAS54*, and *GRAS27* factors of guava fruit, which are involved in the biosynthesis of ethylene in strawberries and tomatoes when its overexpression accelerates ripening [39,40].

FMIR319 was expressed in both the breaker and ripe stages of the guava fruit, with a maximum fold change of 7.09 observed in some of its VS/Ss. These results are consistent with similar studies conducted on pepper (*Capsicum anuum*), where a differential expression between immature and mature stages was also observed [41]. This family has been identified in all maturation stages of climacteric fruits, such as melons, bananas, and tomatoes [18]. miR319 controls six transcription factors, namely, TCP class 2 (TCP2, TCP3, TCP4, TCP9, TCP10, and TCP24) [42]. TCP9 and TCP4 are particularly essential for fruit ripening. During the early phases of ripening in strawberries, for instance, the *TCP9* factor is expressed and plays a crucial role in the modification of the cell wall and the production of anthocyanins and abscisic acid (ABA) [43]. The TCP4/LANCELOATE factor, on the other hand, stimulates the expression of the *LOX2* gene, which biosynthesizes jasmonic acid (JA) and activates the *EIN3* factor gene. These genes and molecules are involved in anthocyanin metabolism, the production of volatile compounds, chlorophyll degradation, and ethylene signaling, respectively [44,45,46]. Their inhibition would result in two situations that would affect ripening; the first is an overexpression of the *EIN3* factor, resulting in an increase in endogenous ethylene production and accumulation. The second effect is a reduction in the production of anthocyanins, volatile compounds, and chlorophyll degradation. Moreover, the TCP4/LANCELOATE factor regulates the *SIYUCCA4* gene involved in the control of auxin transport: an essential phytohormone for fruit ripening [47].

The FMIR393 a/b/c/d isoforms were detected in both stages of the guava fruit. This FMIR has been identified in climacteric fruits, including bananas, grapefruit, and melons [3,30,31]. In these fruits, the highest levels of expression were observed during the breaker and ripe stages [3], and similar results were observed in this study with guava. The target genes encode TIR1/AFBs proteins whose role is to regulate auxin transport via Auxin-Responsive Factors (ARFs) [48]. In melons that overexpressed miRNA393, the maturation time was delayed by up to 7 days without a change in fruit weight, size, or sugar content [4].

Three VS/Ss and one isoform of FMIR395 were found in both stages of guava fruit. The same isoforms and expression profiles were discovered in the blueberry (*Vaccinium angustifolium*) [14]. The target genes of FMIR395 are the *SULTR2:1* of *A. thaliana*, which are expressed in response to sulfate stress, indicating that FMIR395 plays a crucial role in the regulation and assimilation network of sulfates in plants [49]. MiRNA395 overexpression would inhibit sulfurylase enzymes, increasing the concentration of free sulfur in cells. If this is true, excess sulfur would be used in the biosynthesis of methionine, one of the primary precursors to ethylene [49,50].

Isoform 396b of FMIR396 and its six VS/Ss were mostly expressed in the breaker stage with a fold change of 9.67, compared to 1.29 corresponding to the ripe stage, in guava fruit. miR396 has been previously reported to be expressed in the pear and tomato fruits during the breaker and mature stages [19,51,52], confirming our findings. miR396 is involved in the softening of the cell wall by regulating the *βE1,4* gene, which breaks the β-1,4 bonds between hemicellulose residues; increasing the accumulation/generation of carotene in tomatoes [52,53]; and the degradation of sugars and organic acids during pear fruit ripening [51]. Under cold conditions, miRNA396b acts as a direct regulator of *ACO2* in the tomato and trifoliate orange (*Poncirus trifoliata*) fruits [46,54]. It also targets the GRF transcription factors, which are involved in leaf growth and development and the responses to saline and flooding stresses [55].

From the miR397 family, only isoform 397a was found in the ripe stage of the guava fruit, which is consistent with the results found for climacteric-type fruits [18]. miR397 targets the *LAC* genes involved in the biosynthesis of lignin and flavonoids such as anthocyanin in addition to acting as multicopper oxidases, which are responsible for reducing reactive oxygen species to H_2_O [56]. In the mulberry (*Morus atropurpurea*), its overexpression decreases fruit coloration and increases ripening time, whereas its inhibition has the opposite effect. This is because *LAC11a* promotes the expression of genes involved in the biosynthesis of anthocyanins [57], which are required for ripening [58].

In accordance with observations made in blueberry fruit [14], melons [3], olives [15], and Kyoho grapes [11], isoform miR398a of the miR398 family was identified in mature guava fruit. Its target genes are *CSD1* and *CSD2*, which reduce reactive oxygen species (ROS) such as H_2_O_2_ and O_2_^−^ in plants [59], and *CCSD1*, which transports copper ions (Cu^2+^) to *CSD1* and *CSD2* [60]. Since ripening is an oxidative process that requires enzymes to maintain the balance of the ROS produced during respiration [59], the role of FMIR398 during ripening can be divided into two distinct systems; the first is the inhibition of *CSD1/2* in the final stages of ripening, which has been confirmed in tomato fruit [61] and Kyoho grapes [11]. This decrease causes an accumulation of reactive oxygen species (ROS) in the mitochondria of a fruit’s pericarp, resulting in senescence [59,61] and demonstrating the significance of miR398 in the final phases of ripening. The second system involves the accumulation of Cu^2+^ ions as a result of the regulation of the *CCSD1* gene, which is responsible for supplying these ions to *CSD1/2* [60]. Cu^2+^ ions would then be transported by the Antagonist Responder 1 (RAN1) protein to the ETR1 ethylene receptor, which requires Cu^2+^ as a cofactor to detect ethylene, thereby modulating the expression of ripening-related genes [62]. This evidence suggests that members of the miR398 family play an important role in senescence and the production of enzymes related to the final stages of ripening.

In guava fruit, the isoforms a/b and 13 VS/Ss of the FMIR482 were identified. These were only expressed in the breaker stage, similar to what was observed in strawberries [63]. FMIR482 targets NBS-LRRs proteins, which confer resistance to pathogenic attacks such as *Phytophtora infestans* [64]. FMIR482 has been linked to the metabolism of organic acids and sugars in fruits during the final phases of ripening [30] in trifoliate oranges. In tomatoes, it has been found to target the *pectate lyase* genes, which break the bonds between residues of esterified pectin, whose degradation causes fruit softening, as well as the *ZDS* genes, which are involved in carotenoid metabolism and are also essential for ripening [19,53].

In both phases of guava fruit, three FMIR535 VS/Ss and 535a/b/d isoforms were identified. These expression profiles have also been identified in pears, apples, and blueberries [2,14,65]. FMIR535 functions are involved in multiple processes, such as increasing the production of ascorbic acid [34], a metabolite found in high amounts in guava fruit [66]; the negative regulation of the HT1 enzyme, which is responsible for stomatal movements in the presence of CO_2_, light, or phytohormones such as ABA [67]; and the inhibition of ethylene biosynthesis in apples [2], indicating that FMIR535 is indispensable in fruit senescence. In pears, it was found that isoforms 535a/b inhibit the expression of the O_2_-residue-catalyzing *LOX2* gene, thereby promoting the synthesis of hydroperoxide derivatives that contribute to the development of the characteristic aromas and flavors of mature fruits [19,64].

### 3.4. Identification and Ontological Classification of Guava microRNA Target Genes

With the exception of FMIRs 6478, 8155, and 10219, the potential target genes of the guava FMIRs were identified. In total, 80 potential target genes were identified for the 25 unique FMIRs. The FMIR with the greatest number of potential targets was FMIR5139, with 11 possible targets, including *EXPANSIN1* and *βE1,3*, which were classified as gene targets for this FMIR in rose petals during flowering [68] and are also related to the softening of the fruit [52]; the rest of the target genes were derived from multiple alignments made between the miRNA sequence and the guava genome assembly. Next was FMIR164, with eight target genes, including *OMTN* 1-6 in *Oriza sativa* [69] and the *CUC1* and *CUC2* proteins in *A. thaliana* [70]. The remaining families had five or fewer target genes (Figure 3A).

Most of the target genes of the miRNAs found in guava fruit are transcription factors (40%) and genes for responses to phytohormones like ethylene (18%). The GO analysis identified 47 different biological processes, most of them referring to biological, metabolic, and gene expression regulation (GO:0065007, GO:0048519, GO:0010468, GO:0019222); transcription and reproduction processes (GO:0006351, GO:0000003); responses to biotic and abiotic stresses (GO:0009628, GO:0042221, GO:0006950); and the metabolism of phytohormones like ethylene and gibberellin (GO:0009723, GO:0009739, GO:0010033). Similar data have been described for melon, blueberry, and banana fruit, where the majority of found GO terms are related to phytohormone responses, biological/metabolic regulation, and transcriptomic regulation [3,12,14]. These findings suggest that miRNAs expressed during ripening target biological processes like DNA-dependent transcription, stress responses, and biological regulation.

### 3.5. miRNAs Related to Ripening

On the basis of the identification and ontological classification of the target genes of the 28 miRNA families (FMIRs) found in guava fruit, FMIR168, FMIR169, FMIR396, FMIR397, and FMIR482 were identified as directly affecting ripening through processes including ethylene biosynthesis and signaling, the color and fruit ripening time, and the fruit firmness. On the other hand, FMIR160, FMIR165, FMIR167, FMIR393, FMIR395, FMIR398, and FMIR535 influence maturation indirectly due to their association with the regulation of the transport of phytohormones such as ABA, auxins, and gibberellins and the production of volatile compounds. Finally, families FMIR159, FMIR166, FMIR171, FMIR535 and FMIR319 are implicated in both ways (Figure 4 and Table 2). Similar results were found in olive and orange fruits, for which it has been stated that most of the miRNAs founded during the ripening process are related to phytohormone metabolism, cell wall modifications (fruit firmness), ethylene signaling/regulation and sugar degradation [15,30], suggesting that miRNAs expressed during ripening have major roles in some of the most important and characteristic traits of ripe fruits.

## 4. Conclusions

Six libraries of guava fruit cv. Siglo XXI miRNAs expressed during the breaker and ripe phases were constructed. In total, 174 mature miRNA sequences were identified and profiled through ontological and similarity analysis and then categorized into 28 families. From these families, it was determined that miR159, miR169, miR171, miR319, miR396, miR397, and miR482 regulate the ripening process directly (biosynthesis, signaling and production of ethylene, coloration, firmness, and fruit ripening time), whereas miR160, miR165, miR166, miR168, miR393, miR395, miR398, and miR535 are related indirectly (regulation of the transport of phytohormones IAA and ABA, auxins, and gibberellins and generation of volatile compounds). Moreover, it was determined that the majority of miRNA isoforms regulate essential phytohormone metabolisms, such as auxin, gibberellin, anthocyanin, and carotenoid metabolism, not only for the proper development and growth of the plant but also for the fruit.

## Figures and Tables

**Figure 1 genes-14-02029-f001:**
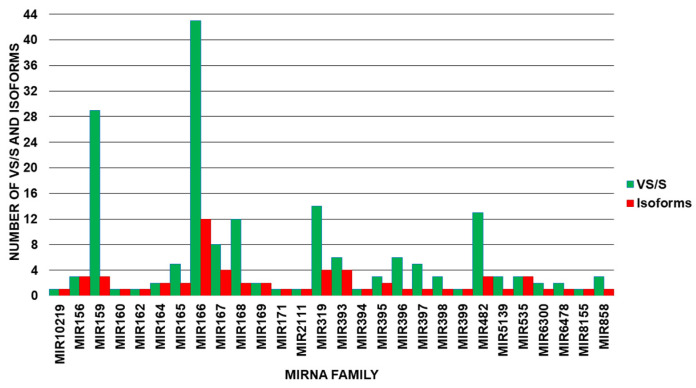
Number of expressed isoforms and VS/Ss per miRNA family.

**Figure 2 genes-14-02029-f002:**
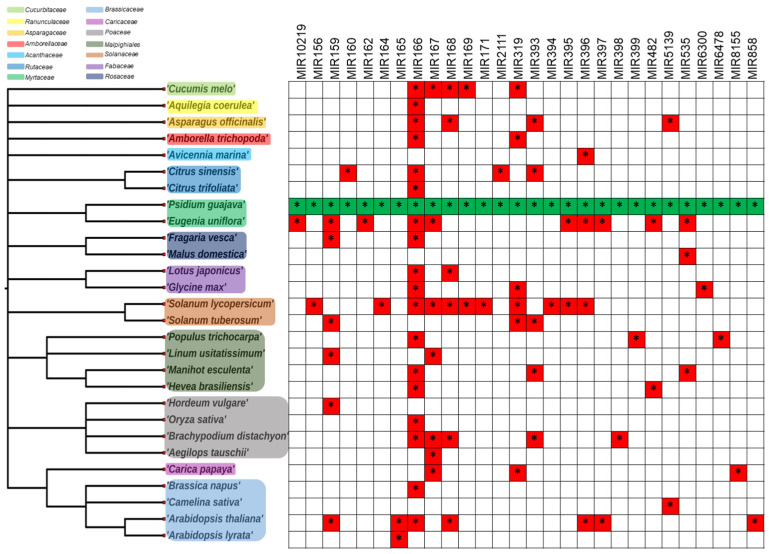
Taxonomic distribution of miRNAs found in *P. guajava* relative to other plant species. On the left panel, a taxonomic tree of 28 plant species from 14 different families is shown. Each miRNA family is depicted at the top. On the right panel, a table of miRNA conservation is displayed; miRNAs found in guava fruit are highlighted in green, while miRNAs found in other plants are highlighted in red. * = Present in plant.

**Figure 3 genes-14-02029-f003:**
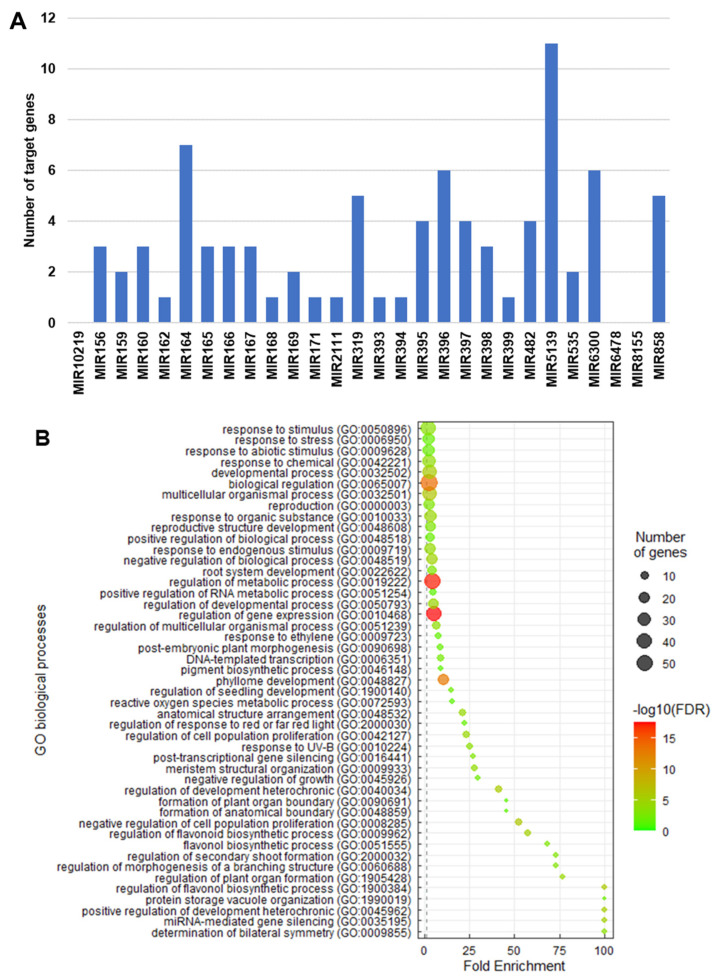
Distribution and ontological enrichment of guava microRNA target genes. (**A**) The number of gene targets per FMIR. (**B**) Ontological enrichment analysis of guava miRNA target genes.

**Figure 4 genes-14-02029-f004:**
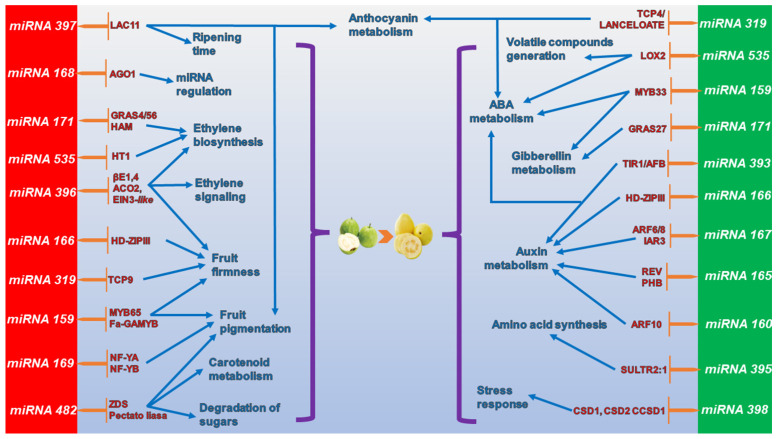
The relationships between miRNAs and their target genes in relation to different ripening processes. MiRNAs directly related to ripening have a red background, while miRNAs indirectly related to ripening have a green background. The processes involved for each target gene are depicted in blue. *LAC11*, *Laccase11*, *AGO1*, *Argonaute1*, *GRAS4/27/56 GIBBERELLIN-INSENSITIVE (GAI)*, *REPRESSOR OF GAI-3 (RGA) AND SCARECROW (SCR)*, *GA*, *gibberellin acid4/27/56*, *HAM*, *Hairy Meristem*, *HT1*, *serine/threonine kinase activity*, *βE1,4*, *β-1,4-endoglucanase1*, *ACO2*, *1-Aminocyclopropane-1-Carboxylic Acid Oxidase2*, *EIN3-like*, *ETHYLENE INTENSIVE3 like transcription factor*, *HD-ZIP III*, *HOMODOMAIN LEUCINE ZIPPER III*, *TCP4/9*, *EOSINTEBRANCHED 1/CYLOIDEA/PROLIFERATIN CELL FACT4/9*, *GAMYB 33/64*, *GIBBERELLIC ACID DEPENDENT MYB33/64*, *NF-YA/B*, *NUCLEAR FACTOR YA/B*, *ZDS*, *zeta-carotene desaturase*, *LOX2*, *LYPOXIGENASE2*, *TIR1/AFB*, *TRANSPORT INHIBITOR RESPONSE1/AUXIN RECEPTOR F-BOX*, *ARF6/8/10*, *AUXINE RESPONSE FACTORS6/8/10*, *IAR3*, *IAA-ALA RESISTANT3*, *REV*, *REUTA*, *PHB*, *PHABULOSA*, *SULTR2:1*, *ATP sulfurylase2:1*, *OG*, *obscuring genes response*, *CSD1/2*, *Copper/Zinc SUPER OXIDE DISMUTASE1/2*, *CCSD1*, and *CHAPERONE Copper/Zinc SUPER OXIDE DISMUTASE1*.

**Table 1 genes-14-02029-t001:** miRNA families, isoforms, and target genes of guava miRNAs.

miRNA Family	VS/S Isoforms	Target Gene	R/M ** Fruit
MIR10219	1/1*	Unknown function	M
MIR156	3/a, b, c	*SQUAMOSA promoter (SPL) 5/8/10*	R/M
MIR159	29/a, b, c	*MYB33*, *MYB65*	R/M
MIR160	1/c	*ARFs 10/16/17*	R/M
MIR162	1/1*	*Dicer-like 1*	R/M
MIR164	2/a, b	*NAC TRANSCRIPTION FACTOR (OMTN1-6)*, *CUP-SHAPED COTYLEDON1/2 (CUC1*, *CUC2)*	R
MIR165	5/a, b	*ATHB-9*, *14*, *15*, *PHAVOLUTA* (*PHV)*, *PHABULOSA* (*PHB)*	R/M
MIR166	43/a, b, c, d, e, f, h, i, k, m, p, u	*ATHB-9*, *14*, *15*, *PHV*, *PHB*	R/M
MIR167	8/a, b, c, f	*ARFs 6*, *8*, *IAA-ALA RESISTANT3* (*IAR3)*	R/M
MIR168	12/a, b	*ARGONAUTE1 (AGO1)*	R/M
MIR169	2/a, g	*NUCLEAR FACTOR YA3* and *5 (NF-YA3*, *NF-YA5)*	R/M
MIR171	1/f	*SCLII-6*	M
MIR2111	1/1*	*F-box/kelch repeat protein (FBK)*	R/M
MIR319	14/a, b, c, q	*TCP 2,3*, *4*, *10*, *24*	R/M
MIR393	6/a, b, c, d	*TIR1/AFB2*	R/M
MIR394	1/1*	*Cyclin-like F-box*	R
MIR395	3/a, b	*ATP-sulfurylases (SULTR2)*, *WRKY26*, *APS1*, *2*	R/M
MIR396	6/b	*1-aminocyclopropane acid oxidase 2 (ACO2)*, *β-1,4 endoglucanase (βE1,4)*, *GROWTH-REGULATION FACTOR* (*GRF)*, *ETHYLENE-INSENSITIVE 3* (*EIN3)*	R
MIR397	5/a	*Laccase (LAC)*, *CBK3 (CK2)*	M
MIR398	3/a	*Copper/zinc superoxide dismutase chaperone 1 (CCSD1)*, *Copper/zinc superoxide dismutase 1 and 2* (*CSD1* and *CSD2) (cytosolic and chloroplastic)*	M
MIR399	1/e	*PHO2*	R/M
MIR482	13/a, b, c	*Pectate Liase*, *ZETA CAROTENE DE-SATURASE* (*ZDS)*, *binding site domains-leucine-rich receptor* (*NBS-LRR)*, *SMG7*	R
MIR5139	3/a	*Expansin1*, *C3HC4/TPX2*, *Gypsy_Ty3-element*, *β-1,3-endoglucanase (βE1,3)*, *Phosphatidylinositol 4-kinase γ 2*, *RRM/RBD/RNP*, *Cyclin-L1-1*, *SCL 7*, *subunit SNAP43*, *Pre-mRNA-splicing factor 38*	R/M
MIR535	3/a, b, d	*Serine/threonine kinase (HT1)*, *LIPOXYGENASE2* (*LOX2)*	R/M
MIR6300	2/1	*PERK2*, *BEE1*, *DAHP1*, *PHYB1*, *HMT*, *HSP70*	M
MIR6478	2/1	Unknown function	R/M
MIR8155	1/1*	Unknown function	R/M
MIR858	3/a	*MYB11*, *MYB111*, *MYB12*, *MYB7*, *MYB48*	R

1/1* = only one isoform obtained, ** R = breaker stage, M = ripe stage, lowercase letters = isoform nomenclature.

**Table 2 genes-14-02029-t002:** miRNAs directly or indirectly related to ripening.

miRNA	Type of Relationship with Ripening	Target Gene	Observations
159a/b/c	Indirect/Direct	*MYB33*, *Fa-GAMYB*, and *MYB65* transcription factors	Firmness and fruit coloration, ABA and gibberellin production
160c	Indirect	*ARF10*	Auxin transport
165a/b	Indirect	*REV* and *PHB* transcription factors	Auxin transport
166a/b/c/d/e/f/h/i/k/m/p/u	Indirect/Direct	*HD-Zip III* protein family	Fruit firmness, auxin transport
167a/b/c/f	Indirect	*ARF6/8* and *IAR3* transcription factors	Auxin transport
168a/b	Direct	*AGO1*	miRNA activity regulation
169a/g	Direct	*NF-YA* and *NF-YB* transcription factors	Fruit coloration
171f	Indirect/Direct	*GRAS4*, *GRAS54*, *GRAS27*, and *HAM* transcription factors	Gibberellin-mediated signaling, ethylene biosynthesis
319a/b/c/q	Indirect/Direct	*TCP9* and *TCP4/LANCELOATE* transcription factors	Fruit firmness, ABA and anthocyanin production
393a/b/c/d	Indirect	*TIR1/AFB*	Auxin transport
395a	Indirect	*SULTR2:1*	Methionine and cysteine biosynthesis
396b	Direct	*βE1,4*, *ACO2*, and *EIN3-like*	Signaling and biosynthesis of ethylene, fruit firmness
397b	Direct	*LAC11*	Fruit coloration, ripening time, anthocyanin biosynthesis
398a	Indirect	*CSD1*, *CSD2*, and *CCSD1*	Metallic-induced stress and ROS regulation
482c	Direct	*Pectate liase* and *ZDS*	Carotenoid metabolism, fruit firmness, sugar degradation
535a/b/d	Indirect	*HT1* and Lipoxygenase2 *LOX2*	ABA-signaling, ethylene biosynthesis, volatile compound generation

lowercase letters = isoform nomenclature.

## Data Availability

Data are contained in the article and Appendix A.

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
