# Peer review of "Identification and Profiling Analysis of microRNAs in Guava Fruit (Psidium guajava L.) and Their Role during Ripening"

_genes, 2023, doi:10.3390/genes14112029_

Round 1

Reviewer 1 Report

Comments and Suggestions for Authors

Comments, 

1)    Significance of study is missing in abstract.

2)    Line 16. Two miRNA libraries? Did you take enough replicates for statistical analyses?

3)    Line 118. Citation for tool is missing. Is it right name of tool.

4)    Which tool used for mapping? Lot of information on miRNA seq data analyses is missing. Please consider re-writing.

5)    Line 114-126. Consider re-writing.

6)    Line 129. Did you take enough replicates? Just two libraries?

7)    Authors need to submit the raw sequencing reads to NCBI-SRA. Its mandatory step before publication.

8)    Check language of Table 1. Is it English? Please re-check.

Main issues with paper are- Not enough replicates, authors should have at least 3 replicates per sample to have enough statistical power. The data analysis methodology is not clear. Also, language is issue. At some instances language other than English has been used. I will highly recommend re-writing and resubmitting the paper.

Comments on the Quality of English Language

Also, language is issue. At some instances language other than English has been used. I will highly recommend re-writing and resubmitting the paper.

Author Response

Response to Reviewer 1 Comments:

1. Summary

Thank you very much for taking the time to review this manuscript. Please find the detailed responses below and the corresponding revisions highlighted/in track changes in the re-submitted files.

2. Questions for General Evaluation

Reviewer’s Evaluation

Response and Revisions

Does the introduction provide sufficient background and include all relevant references?

Must be improved

Improved

Are all the cited references relevant to the research?

Must be improved

Improved

Is the research design appropriate?

Must be improved

Improved

Are the methods adequately described?

Must be improved

Improved

Are the results clearly presented?

Must be improved

Improved

Are the conclusions supported by the results?

Must be improved

Improved

3. Point-by-point response to Comments and Suggestions for Authors

Comments 1: Significance of study is missing in abstract.

Response 1: Thank you for pointing this out. We agree with this comment. Therefore, we have included the importance of this study regarding the understanding of the regulatory process of the genes involved in the ripening processes.

Comments 2: Line 16. Two miRNA libraries? Did you take enough replicates for statistical analyses?

Response 2: Each library was constructed with 3 replicas and sequenced by duplicate, you cand find more information about it in the Materials and Methods section.

Comments 3: Line 118. Citation for tool is missing. Is it right name of tool.

Response 3: Line 124. Citation and function name are right for the QIAGEN CLC Genomics Workbench, according with the website of the developer.

Comments 4: Which tool used for mapping? Lot of information on miRNA seq data analyses is missing. Please consider re-writing.

Response 4: We used the Maps read to reference function of the QIAGEN CLC Genomics Workbench. Line 118. Sequencing and bioinformatic analysis of miRNA libraries 2.5. section has been re-written and improved.

Comments 5: Line 114-126. Consider re-writing.

Response 5: Lines 119-140 has been re-written.

Comments 6: Line 129. Did you take enough replicates? Just two libraries?

Response 6: Line 129 has been re-written and now it specifies that samples were divided into two groups: ripe stage and breaker stage, each group with three repetitions, and each repetition was sequenced by duplicate, ending with twelve different reads and six miRNA libraries.

Comments 7: Authors need to submit the raw sequencing reads to NCBI-SRA. Its mandatory step before publication.

Response 7: Raw sequences of the miRNA-seq, has been already submitted to the NCBI under the accession number PRJNA996171. This information was added in a new section called 2.6. Availability of data and materials in Line 142.

Comments 8: Check language of Table 1. Is it English? Please re-check.

Response 8: Thank you for noticing, Table 1 language was fixed to be English.

4. Response to Comments on the Quality of English Language

Response 1: Thank you for your comments on the quality of the English language, we already have re-written the entire paper in order to improve this problem.

5. Additional clarifications

Main issues stated by this reviewer were already fixed, which includes the clarification of how many replicates were done, and improving the writing in the data analysis methodology. Also, the language was improved to be more understandable.

Reviewer 2 Report

Comments and Suggestions for Authors

Major comments:

The analysis of microRNAs expression during the ripening process of guava, a climacteric fruit, is a very interesting topic. However, the submitted paper does not provide the basic descriptions of the method and Supplementary data, making adequate peer review impossible. Therefore, I request that significant additions and revisions of the descriptions be made.

Minor comments:

L17: What does "media China" mean?

L27: "guava" is required for keywords.

L47: aroused → arisen

L62: AIAIAA

L66: "midia" or "media"?

L78: Does the sample include pericarp and seeds?

L78-79: Describe the RNA extraction method.

L81: gr?

L83: What is "photo-documenter"?

L106-109: Font size of alphabet is different from others.

L113: section 2.3. section2.2.

L115-116:A detailed description of the miRNA sequencing is needed.

L129:"Each with three repetitions" should be described in "2. Materials and methods".

L133-134, Figure1:What are the sub and super variants?  How do you understand its existence?

L135:Attach "Supplementary data1".

L142-144: Describe the frequency distribution in a diagram.

Figure1: Members isoforms

Table1: RIMFs → MIRs

Table1: English the Spanish words, "Familia", "isoformas", "Genes objetivo", and "Fruto".

Table1: VS/S/isoformas* → 1/1* 

Table1: MIR159  Delete "y" in "MYB33 y".

Table1: MIR167  Delete "y" in "ARFs 6,8 y"

Table1: In Table 1 and Figure 1, there are families (MIR156, 159, 164, 165, 167, 168, 319, 396, 5139, and 535) whose data do not match.

L147: 3.1. 3.2.

Figure2.: Describe the method of the phylogenetic analysis.

Is it impossible to create a phylogenetic tree because of the lack of information?

Figure2.: 27 plant species → 28 plant species

Figure2.: 16 different families → 14 different families

L167: Attach TableS1 and TableS2.

L228: FMIR and MiRNA are used without distinction.

L235: Show the data.

L283: Provide a citation regarding the trifoliate orange.

L347: " using the R statistical package[66] " should be described in "2. Materials and methods".

Table2: ehtylene → ethylene

L398: AIAIAA

L417-426: Put the Spanish sentence into English.

Comments on the Quality of English Language

Needs to be rechecked by native speakers.

Author Response

Response to Reviewer 2 Comments

1. Summary

2. Questions for General Evaluation

Reviewer’s Evaluation

Response and Revisions

Does the introduction provide sufficient background and include all relevant references?

Yes

Are all the cited references relevant to the research?

Yes

Is the research design appropriate?

Yes

Are the methods adequately described?

Must be improved

Improved

Are the results clearly presented?

Must be improved

Improved

Are the conclusions supported by the results?

Yes

3. Point-by-point response to Comments and Suggestions for Authors

Comments 1: The analysis of microRNAs expression during the ripening process of guava, a climacteric fruit, is a very interesting topic. However, the submitted paper does not provide the basic descriptions of the method and Supplementary data, making adequate peer review impossible. Therefore, I request that significant additions and revisions of the descriptions be made.

Response 1: Thank you for pointing this out. We agree with this comment. Therefore, we have included the corresponding Supplementary data and figures, which include detailed information about raw miRNA sequences, fold change values and ranges of distribution of the sequences.

Comments 2: L17: What does "media China" mean?

Response 2: Line 76: It refers to the group/type or variety of a Mexican guava fruit, very common in this region.

Comments 3: L27: "guava" is required for keywords.

Response 3: Line 29: included

Comments 4: L47: aroused arisen

Response 4: Line 50: corrected

Comments 5: L62: AIAIAA

Response 5: L65: corrected

Comments 6: L66: "midia" or "media"?

Response 6: Line 69: corrected to Media

Comments 7: L78: Does the sample include pericarp and seeds?

Response 7: Line 82: Corrected, we used only the pericarp of the fruit

Comments 8: L78-79: Describe the RNA extraction method.

Response 8: Line 82-83. We used the CTAB method, citation is now included.

Comments 9: L81: gr?

Response 9: Line 85: corrected to g

Comments 10: What is "photo-documenter"?

Response 10: Line 87: we wanted address an UV transilluminator, corrected.

Comments 11: L106-109: Font size of alphabet is different from others.

Response 11: L110-113: Thank you for noticing, corrected.

Comments 12: L113: section 2.3. section2.2.

Response 12: Line 117: corrected, thank you.

Comments 13: L115-116:A detailed description of the miRNA sequencing is needed.

Response 13: Line 119-140. We have included the sequencer used, and also the quantity of RNA used for each sample. Furthermore section 2.5 was re-written.

Comments 14: L129:"Each with three repetitions" should be described in "2. Materials and methods".

Response 14: Line 119-120: Moved to the right section and re-written for better understanding.

Comments 15: L27: L133-134, Figure1:What are the sub and super variants?  How do you understand its existence?

Response 15: L153-155: Explained as absence or addition nucleotides at any end of the miRNA sequence.

Comments 16: L135:Attach "Supplementary data1".

Response 16: Already submitted.

Comments 17: L142-144: Describe the frequency distribution in a diagram.

Response 17: Included in the Supplementary Material 1 under the name and description of: Supplementary Figure 1. Frequency distribution of sizes of miRNAs found in guava

Comments 18: Figure1: Members isoforms

Response 18: Has been now changed to isoforms

Comments 19: Table1: RIMFs → MIRs

Response 19: Corrected

Comments 20: Table1: English the Spanish words, "Familia", "isoformas", "Genes objetivo", and "Fruto".

Response 21: Table 1. Has been corrected, thank you for noticing.

Comments 22: Table1: VS/S/isoformas* → 1/1* 

Response 22: Corrected

Comments 23: Table1: MIR159 Delete "y" in "MYB33 y".

Response 23: Deleted

Comments 24: Table1: MIR167 Delete "y" in "ARFs 6,8 y"

Response 24: Deleted

Comments 25: Table1: In Table 1 and Figure 1, there are families (MIR156, 159, 164, 165, 167, 168, 319, 396, 5139, and 535) whose data do not match.

Response 25: Table has been updated with the right values

Comments 26: L147: 3.1. 3.2.

Response 26: Line 169. Corrected, thank you

Comments 27: Figure2.: Describe the method of the phylogenetic analysis.

Is it impossible to create a phylogenetic tree because of the lack of information?

Response 27: We agree, it is not a phylogenetic analysis but instead a Taxonomic distribution of miRNA conservation in other plants. Has been corrected in all the document.

Comments 28: Figure2.: 27 plant species  28 plant species

Response 28: Corrected

Comments 29: Figure2.: 16 different families  14 different families

Response 29: Corrected

Comments 30: L167: Attach TableS1 and TableS2.

Response 30: Already submitted inside Supplementary Material 1 and 2

Comments 31: L228: FMIR and MiRNA are used without distinction.

Response 31: Replaced miRNA for FMIR in the right sections of the document

Comments 32: L235: Show the data.

Response 32: Line 254: This was a multiple alignment and BLASTn analysis performed only for this miRNA, so including it will mean to add 2 additional figures for only one result. Supplementary Material 1 includes the complete mature sequence for miRNA171f so everyone can do it for himself.

L283: Provide a citation regarding the trifoliate orange.

Response 33: Line 304: a citation has been now added under the number [54]

Comments 34: L347: " using the R statistical package [66] " should be described in "2. Materials and methods".

Response 34: Line 138-139: This part has now been moved to the right section

Comments 35: Table2: ehtylene  ethylene

Response 35: Corrected, thank you

Comments 36: L398: AIAIAA

Response 36: Line 415: corrected.

Comments 37: L417-426: Put the Spanish sentence into English.

Response 37: Line 435-439: translated to English, thank you.

4. Response to Comments on the Quality of English Language

Point 1: Needs to be rechecked by native speakers.

Response 1: Thank you for your comments on the quality of the English language, we already have re-written the entire paper in order to improve this problem.

Reviewer 3 Report

Comments and Suggestions for Authors

In this manuscript, the authors analyse microRNAs and their roles in the ripening of Mexican guava fruit. The authors very nicely use an array of bioinformatic methods and assays to show that 174 identified miRNAs belong to 28 different families. Moreover, MiRNA families and their target genes are well discussed. The miRNA families MIR168, MIR169, MIR396, MIR397 and MIR482 are found as directly related to guava fruit ripening. These discoveries are expected to have an interesting impact in the understanding of the genes involved in the guava fruit ripening process and its gene regulation. Moreover, the work exhibits proficient writing skills and is very accessible for readers. I only have some minor requests for revision.

In the Plant Material part, the authors collected fruit samples of different stages from a single tree. I am wondering which part of the fruit are used for RNA extraction? Please give more details.

Author Response

Response to Reviewer 3 Comments

1. Summary

2. Questions for General Evaluation

Reviewer’s Evaluation

Response and Revisions

Does the introduction provide sufficient background and include all relevant references?

Can be improved

Improved

Are all the cited references relevant to the research?

Yes

Is the research design appropriate?

Can be improved

Improved

Are the methods adequately described?

Yes

Are the results clearly presented?

Yes

Are the conclusions supported by the results?

Yes

3. Point-by-point response to Comments and Suggestions for Authors

Comments 1: In the Plant Material part, the authors collected fruit samples of different stages from a single tree. I am wondering which part of the fruit are used for RNA extraction? Please give more details.

Response 1: Thank you for pointing this out. We agree with this comment. Therefore, we have included that we used only the pericarp of the fruit. Furthermore, Materials and Methods section, now include specific details about samples and methods used in this work, also has been re-written for better understanding.

Round 2

Reviewer 2 Report

Comments and Suggestions for Authors

Compared to the previous paper, this paper is much easier to understand due to significant improvements. However, some modifications are needed.

L28: The key word "climacteric fruit" is more appropriate than "ethylene". Correspondingly, change "fruit ripening" to "ripening".

L45: Delete "-" in "entering-the".

L78: Delete "-" in "of-guava".

L105: sup → sub

L106: sub/sup  → sub/super

L107: VS/V  → VS/S

L143: The title of table1: miRNA families, isoforms and target genes of guava miRNAs

L144: state → stage,   state → stage

Table 1   VS/S* isoforms  →  1/1* (in MIR10219 line)

Table1: In Table 1 and Figure 1, there is a Family (MIR 156, 164) whose data do not match.

L148: In Fig. 2, FMIR167 is not present in Cucumis melo. This is contradictory to the statement in the text.

L163: 3.1.  3.3.

L186: The statement, this family is one of the most conserved in plants, is inconsistent with the description in Fig. 2.

L192: Are citations 25, 26 references to climacteric fruit?

L224: a/b a/g

L236: ripe and breaker stages → breaker and ripe stages

L314: Are citations 64 references to tobacco mosaic virus?

L335: FMIR5139 with 11 possible targets?  Recheck.

L337: 7 target genes?  Recheck.

L355: 3.4.  3.5.

L355: The 3.5. section needs discussion.

L363: FMIR 535 is missing from the description.

L394: Delete "-" in "antho-cyanin".

L395: for the produce?

L398: FC?

Supplementary Figure1:  guava guava fruit

Indicate "Table S2" in Table S2.

Comments on the Quality of English Language

English language has improved.

Author Response

For research article: Identification and profiling of microRNAs in two ripening stages of guava fruit (Psidium guajava L.)

Response to Reviewer 1 Comments:

1. Summary

Thank you very much for taking the time to review this manuscript. Please find the detailed responses below and the corresponding revisions highlighted/in track changes in the re-submitted files. All the changes are highlighted in red in the main text.

2. Questions for General Evaluation

Reviewer’s Evaluation

Response and Revisions

Does the introduction provide sufficient background and include all relevant references?

Yes

Are all the cited references relevant to the research?

Can be improved

Improved

Is the research design appropriate?

Yes

Are the methods adequately described?

Yes

Are the results clearly presented?

Yes

Are the conclusions supported by the results?

Yes

3. Point-by-point response to Comments and Suggestions for Authors

Comments 1: L28: The key word "climacteric fruit" is more appropriate than "ethylene". Correspondingly, change "fruit ripening" to "ripening".

Response 1: L:28 Changed to climacteric fruit and ripening in keywords

Comments 2: L45: Delete "-" in "entering-the".

Response 2: L:45 Deleted

Comments 3: L78: Delete "-" in "of-guava".

Response 3: Line 78: Deleted

Comments 4: L105: sup sub

Response 4: L105: changed to only sup meaning super

Comments 5: L106: sub/sup sub/super

Response 5: L106: changed to sub/sup, according to the L105

Comments 6: L107: VS/V VS/S

Response 6: L107: changed to VS/S

Comments 7: L143: The title of table1: miRNA families, isoforms and target genes of guava miRNAs

Response 7: L143: Title has been changed

Comments 8: L144: state stage, state stage

Response 8: Thank you for noticing, changed both to stage

Comments 9: Table 1 VS/S* isoforms 1/1*(in MIR10219 line)

Response 9: Table 1: changed in MIR10219, 162, 2111, 394 and 8155 lines

Comments 10: Table1: In Table 1 and Figure 1, there is a Family (MIR 156, 164) whose data do not match.

Response 10: Table 1: Thank you for pointing it out, values in Figure 1 have been fixed to match those of Table 1.

Comments 11: L148: In Fig. 2, FMIR167 is not present in Cucumis melo. This is contradictory to the statement in the text.

Response 11: This has been corrected in Fig 2 according to the text.

Comments 12: L163: 3.1. 3.3.

Response 12: Line 163: Changed

Comments 13: L186: The statement, this family is one of the most conserved in plants, is inconsistent with the description in Fig. 2.

Response 13: Line 186. Removed the sentence, it was mistaken for FMIR166

Comments 14: L192: Are citations 25, 26 references to climacteric fruit?

Response 14: Line 187-192: The references were put in the correct order, thank you for noticing.

Comments 15: L224: a/b a/g

Response 15: L223: already changed

Comments 16: L236: ripe and breaker stages breaker and ripe stages

Response 16: L235: Changed

Comments 17: L314: Are citations 64 references to tobacco mosaic virus?

Response 17: L313: Changed to Phytoptora infestans, which was the right microorganism’s name, thank you.

Comments 18: L335: FMIR5139 with 11 possible targets? Recheck.

Response 18: L334-336: 9 possible target genes were obtained from multiple alignment analyses, and the remaining 2 were from reference number 68.

Comments 19: L337: 7 target genes?  Recheck.

Response 19: L336-338: Reference (69) was missing, in this it is stated that FMIR 164 has these target genes, thank you for bring it to our attention

Comments 20: L355: 3.4. 3.5.

Response 21: L355: Changed, thank you for noticing.

Comments 22: L355: The 3.5. section needs discussion.

Response 22: L364-369: discussion has been added.

Comments 23: L363: FMIR 535 is missing from the description.

Response 23: L363: Added

Comments 24: L394: Delete "-" in "antho-cyanin".

Response 24: L399: Deleted

Comments 25: L395: for the produce?

Response 25: L400: changed to fruit

Comments 26: L398: FC?

Response 26: Line 403: added (Fold Change).

Comments 27: Supplementary Figure1:  guava guava fruit

Response 27: Supplementary Figure 1: Changed

Comments 28: Indicate "Table S2" in Table S2.

Response 28: Corrected